# Glacial–Interglacial Cycles and Early Human Evolution in China

Zhenyu Qin  and Xuefeng Sun *

School of Geography and Ocean Science, Nanjing University, Nanjing 210023, China;
mg21270019@smail.nju.edu.cn
* Correspondence: xuefeng@nju.edu.cn

**Abstract:** China is a crucial region for investigating the relationship between climate change and hominin evolution across diverse terrestrial ecosystems. With the continuous development of palaeo-climatology, chronology, and archaeology, the environmental and hominin record of the Early and Middle Pleistocene in China is steadily accumulating, shedding light on the effects of climate change on the distribution of early human settlements and population dynamics. However, the migration and dispersal of these early humans within long-term climate fluctuations and their underlying mechanisms remain to be clarified. Based on the spatial-temporal distribution of 95 Early to Middle Pleistocene archaeological sites in China, we found that intensified hominin activities gradually shifted southward under the influence of multiple glacial–interglacial cycles. The frequent bidi-rectional movements of these early humans between north and south were assumed as follows. During glacial periods, hominins living in North China migrated to southern areas, while inter-glacial periods witnessed the northward expansion of hominins inhabiting South China. Among all the potential driving mechanisms, we suggest that the available resources in terrestrial ecosystems may be the most fundamental factor. Combined with paleoenvironmental and archaeological records, we provide an Asian perspective for a better understanding of how the glacial–interglacial cycles shaped early human evolution.

**Keywords:** glacial–interglacial cycles; climate change; migration and dispersal; hominin evolution; terrestrial ecosystem; East Asia; Early and Middle Pleistocene

## 1. Introduction

The adaptation to climatic change has played a significant role in the evolutionary history of hominins [1–3]. The Pleistocene era, characterized by extensive glaciation, witnessed substantial shifts in the Earth's climate [4–6]. The advance and retreat of glaciers resulted in repeated alternations between glacial periods and warm interglacial periods, referred to as glacial–interglacial cycles [7,8]. Such cyclic fluctuations in climate had a profound influence on hominin activities across various regions of the Eastern Hemisphere, e.g., [9–13]. The investigation of the impact of climate change contributes to a deeper comprehension of its underlying mechanisms and helps us adjust socio-natural systems [14,15]. Since the last century, scholars have analyzed the relationship between climate and hominin evolution in Africa from different perspectives and proposed a variety of hypotheses, e.g., [16–24]. In Europe, population dynamics during the Middle Pleistocene were vividly described by models such as "ebb and flow" and "source and sink" [25–28]. However, hypotheses about climate–evolutionary linkages for Asia are relatively limited.

Asia is a key area for investigating the correlation between early human activity and climate change [29–32]. The abundant and continuous Quaternary deposits (e.g., loess–paleosol sequences, fluvial–lacustrine deposits) in China not only preserve large numbers of archaeological sites but also provide high-quality paleoenvironmental data [33,34]. With the advancement of archaeological excavations and dating techniques, the understanding

of the climatic impact on hominin activity in China is gradually advancing. For instance, Lu et al. discussed the interaction of the Earth's surface processes with early human occupation [35]. Sun et al. discussed the relationship between hominin activities and glacial–interglacial climate change in the Qinling Mountains region [36]. Yang et al. examined the distribution of archaeological sites during the Early and Middle Pleistocene in China to understand the long-term impact of geography and behavior on hominin populations before, during, and after critical climatic events [37,38]. Recently, increasing evidence has demonstrated a southward migration and northward expansion of the hominin population during long-term climatic fluctuations, e.g., [36–40]. Nevertheless, most of these studies did not elucidate the driving mechanisms of population dynamics. Although some scholars summarized the spatio-temporal distribution of early human settlements, they did not explain the fundamental reason for this phenomenon in detail, e.g., [39,40].

Migration and dispersal are both vital adaptive strategies for seeking suitable habitats, serving as pivotal links in the process of hominin evolution [41–43]. Given the close relationship between population dynamics and cultural innovation, genetic diversity, and cognitive improvement [44–46], understanding the spatial-temporal distribution of hominin settlements is of great concern to archaeologists, geneticists, and paleoanthropologists, e.g., [37,39,47–49]. In this paper, we aim to synthesize the movement patterns of early humans in China during long-term glacial–interglacial climate change and discuss the potential driving mechanisms.

## 2. Materials and Methods

### 2.1. Geographical and Archaeological Settings

China (73°33′–135°05′ E, 3°51′ N–53°33′ N) is located in East Asia and spans nearly 50 degrees of latitude from north to south, with the climatic zones from south to north being tropical, subtropical, temperate, and cold temperate, respectively [50]. China is greatly influenced by the alternating summer and winter monsoons, making it the region with the most typical and significant monsoon climate worldwide, e.g., [51,52]. In summer, the warm and humid East Asian Summer Monsoon (EASM) and the India Summer Monsoon (ISM) blow from the low-latitude ocean to the interior of the Asian continent, bringing abundant monsoonal rainfall to China [53]. In winter, the dry and cold East Asian winter monsoon (EAWM) transports dust from the Asian deserts to East Asia and the North Pacific [54].

The Qinling Mountains, located at about 32° N–35° N, are the natural boundary between the north and south of China in terms of geography, climate, flora, and fauna [55,56]. At present, the northern region of the Qinling Mountains is characterized by a temperate monsoon and a temperate continental climate, encompassing semi-humid, semi-arid, and arid areas [50]. The annual precipitation in these areas usually does not exceed 800 mm, while the winter temperature remains below the freezing point (0 °C) [50]. The vegetation of northern China is dominated by warm temperate deciduous broad-leaved forest and steppe, and the fauna is represented by open grassland species [57]. The southern region of the Qinling Mountains is characterized by a subtropical monsoon climate, featuring annual precipitation exceeding 800 mm and winter temperatures above 0 °C [50]. The vegetation of southern China is dominated by tropical and subtropical evergreen broad-leaved forests [50]. Plant food is available all year round to sustain both hominins and herbivorous animals [57]. The fauna is represented by forest-dwelling taxa adapted to warmth and humidity [58].

Under the influence of the collision of the Eurasian and Indian plates in the early Paleocene and the uplift of the Tibetan Plateau in the Late Cenozoic, China exhibits a geomorphological pattern with a gradual decrease from west to east, presenting as the "Three Gradient Terrains" [59–61]. The first terrain stretches across the Tibetan Plateau in southwest China, with an average elevation of over 4000 m; the second terrain crosses the Kunlun and Qilian Mountains from the Tibetan Plateau to the north and the Hengduan Mountains to the east, with an average elevation of 1000–2000 m. The third terrain is mainly

characterized by the topography of interspersed hills, low mountains, and plains, with an average elevation of less than 500 m [35].

The Early and Middle Pleistocene (~2.4–0.3 Ma, Ma = million years ago) is the period of the long-term evolution of *Homo erectus* and the existence of archaic *Homo sapiens* in East Asia [31,62], such as *Homo erectus* from Yuanmou, Gongwangling, Yunxian, Zhoukoudian, Hexian, and archaic *Homo sapiens* from Dali, Maba, and Panxiandadong Cave [63–71]. Early humans described in this paper fall into both of these lineages. The Chinese *Homo erectus* group has complex physical characteristics and great internal variations and has shown north–south differences in morphological traits [31]. It has been suggested that hominins underwent complex population exchanges, subsistence activities, and environmental adaptations in the late Middle Pleistocene, leading to a diverse evolutionary pattern [72]. According to previous statistics [40], the 95 Paleolithic and hominin fossil sites with published ages during this period are mainly located on the second terrain of China (Figure 1). These Paleolithic sites are clustered in four early human gathering areas, including the Nihewan Basin and the adjacent Zhoukoudian (NHW-ZKD), the Qinling Mountains Range (QMR), the lower reaches of the Yangtze River Valley (YRV), and the south of the YRV (southern China) [40].

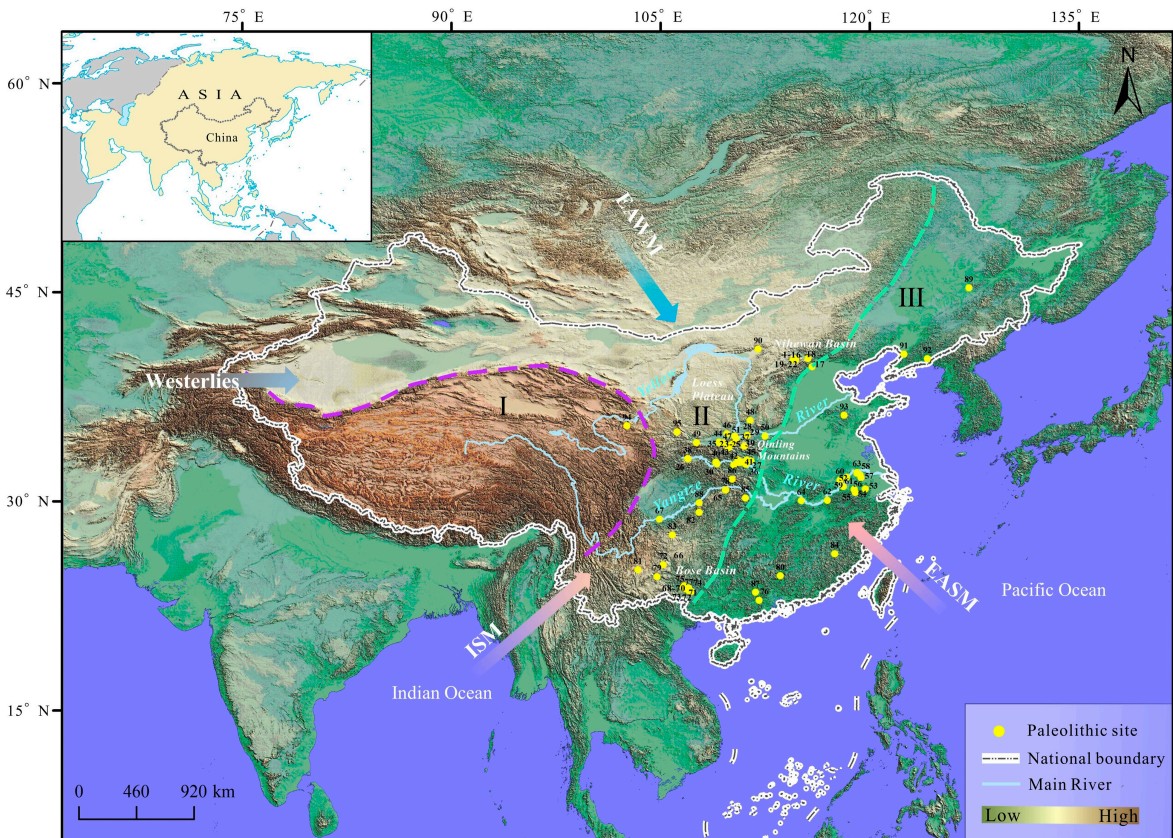

**Figure 1.** Geographical distribution of 95 archaeological sites from the Early and Middle Pleistocene with numerical age estimates in China [40]. Note: the dashed purple line is the boundary between the first (I) and second terrain (II), and the dashed green line is the boundary between the second and third terrain (III). EASM: East Asian Summer Monsoon; EAWM: East Asian Winter Monsoon; ISM: India Summer Monsoon. The 95 archaeological sites are listed as follows: 1. Shanshenmiaozui; 2. Shangshazui; 3. Putaoyuan; 4. Shigou; 5. Lanpo; 6. Majuangou; 7. Dachangliang; 8. Xiaochangliang; 9. Banshan; 10. Nanshanbian; 11. Feilang; 12. Madigou; 13. Cenjiawan; 14. Donggutuo; 15. Huojiadi; 16. Maliang; 17. Zhoukoudian locality-1; 18. Zhuwobao; 19. Hougou; 20. Dongpo; 21. Motianling; 22. Queergou; 23. Shangchen; 24. Gongwangling; 25. Xihoudu; 26. Longgangsi; 27. Meipu; 28. Guanmenyan; 29. Yunxian Man; 30. Yuelianghu; 31. Shangbaichuan; 32. Bailongdong Cave; 33. Miaokou; 34. Chenjiawo;

35. Liuwan; 36. Shuangshu; 37. Qiaojiayao; 38. Yaochangwan; 39. Jiuchang; 40. Luojiacun; 41. Wujiagou; 42. Pengjiahe; 43. Longyadong Cave; 44. Jijiawan; 45. Maling 2A; 46. Dali Man; 47. Zhoupo; 48. Dingcun; 49. Yujiashan; 50. Beiyao; 51. Yuling Man; 52. Renzidong Cave; 53. Qiliting; 54. Chenshan; 55. Yangshan; 56. Maozhushan; 57. Heshangdun; 58. Fangniushan; 59. Hexian Man; 60. Chaoxian Man; 61. Nanjing Man; 62. Hualong Cave; 63. Lianhuadong Cave; 64. Shilongtou; 65. Longgupo; 66. Mohui Cave; 67. Yuanmou Man; 68. Fengshudao; 69. Yangwu; 70. Baigu; 71. Nanbanshan; 72. Nalai; 73. Baifeng; 74. Gaolingpo; 75. Liuhuaishan; 76. Panlong Cave; 77. Baidu; 78. Yumi Cave; 79. Panxiandadong Cave; 80. Maba Man; 81. Zhangkou Cave; 82. Yanhui Cave; 83. Xiaohui Cave; 84. Lingfeng Cave; 85. Changyang Man; 86. Xinglong Cave; 87. Dongzhongyan; 88. Ranjialukou; 89. Jiaojiedong; 90. Dayao; 91. Jinniushan locality A; 92. Miaohoushan; 93. Yiyuan Man; 94. Baishiya Cave; 95. Yangshang.

### 2.2. Glacial–Interglacial Climatic Changes

Since around 2.7 Ma, the global climate has witnessed dramatic glacial–interglacial cycles on a multi-millennial time scale as glaciers expanded dramatically at high latitudes in the Northern Hemisphere [73,74]. Records of the global monsoon system, high-latitude ice volume, global sea level, and deep ocean temperature all exhibited distinct glacial–interglacial cyclicity [75–79]. During warm and wet interglacial periods, the EASM strengthens and the EAWM weakens, while the opposite occurs during cold and dry glacial periods [51]. The temperature variations between warm and cold cycles could reach an average of 5 °C and even 15 °C in the high latitudes of the northern and southern hemispheres [80,81]. The periodic variation of solar radiation influenced by Earth orbit parameters may be the fundamental driving factor for climatic cycles [82].

A global cooling trend was observed around 1.8–1.2 Ma by paleoclimatic reconstruction, with the temperature cooling gradually by approximately 0.34 °C per 100 kyr [83,84]. Loess–paleosol sequences and pollen records from northern China collectively reflected the increasingly arid and cooling climate of inland Asia [85–88]. During the Mid-Pleistocene Transition (MPT, ~1.2–0.7 Ma), the dominant periodicity of the global climate system gradually shifted from 41 ka to 100 ka, resulting in an intensified contrast between glacial and interglacial periods [89–92]. Major changes in the Earth's climate system include the expansion of ice sheets, the intensification of the EAWM, and the attenuation of the EASM [93,94]. The longer duration of harsh glacial periods caused by MPT may pose a great threat to the hominin population in northern China [36–38]. The Mid-Brunhes Event (MBE, ~0.5–0.3 Ma) was another significant global climate change in the Middle Pleistocene and featured enhanced warmth during interglacial periods, further resulting in larger-amplitude glacial–interglacial oscillations, e.g., [95–97]. It is suggested that significant climate fluctuations during the MBE may have played a role in facilitating mid-Pleistocene hominin and behavioral diversity, which could have led to human expansion across wider regions of Eurasia and the extinction or prolonged existence of older lineages, as well as the origin of new species such as "China archaics" in East Asia [97].

Different regions in China made different responses to the glacial–interglacial climatic fluctuations during the Early and Middle Pleistocene. The climate changes observed in the Nihewan Basin over a prolonged period indicated an increasing trend towards aridity, heightened disparities between glacial and interglacial periods, as well as extended durations of harsh cold and dry conditions [98]. For example, a combined mineral-magnetic and geochemical investigation of the Xiantai fluvial–lacustrine sequence revealed a long-term decrease in chemical weathering intensity and consequent increase in aridification in the Nihewan Basin since the Early Pleistocene, providing a general paleoclimate context for early human adaptation in northern China [99]. Loess–paleosol records in western Loess Platea suggest a long-term drying trend since ~1.0 Ma, with two significant abrupt drying events at ~0.65 Ma and ~0.3 Ma [100]. Based on the rainfall estimates over the past 1.1 Ma at the Xifeng section in the central Loess Plateau [101,102], there could have been a potential increase of up to 80% in rainfall levels within the semi-arid western region, while the more humid southern and eastern areas might experience a moderate rise of ~20%

in the interglacial period [98,101,102]. Conversely, in glacial periods, precipitation may decline by ~25% across the plateau, resulting in reduced rainfall ranging from 400–600 mm instead of the previous 600–800 mm [98,101,102]. Under extremely cold and arid conditions, the estimated mean annual precipitation (MAP) was merely 150–250 mm, while the mean annual temperature (MAT) ranged from ~1.5 to 3 °C in the Nihewan Basin [103]. During hospitable periods, the MAT increased by 4–6 °C and the MAP increased by 200–300 mm, resulting in a climate similar to that of the southern YRV today [103]. According to the quantitative climatic reconstruction by Da et al. [96], the MAP was 530 mm and the MAT was 7.9 °C during the warmest/wettest period of the MBE.

In southern China, the red earth sediments contain abundant paleoclimatic information, e.g., [104–107]. In the lower reaches of YRV, phytolith analysis of the Xuancheng profile suggested that frequent activities of Early Pleistocene humans often took place in dry and cool periods during times of milder climates [104]. Ten paleoclimatic wetter periods were recorded by the Jingxian red clay mainly during glaciations since about 0.84 Ma, indicating the regional precipitation should be negatively related to the EASM intensity [105], and the modern anti-phase of monsoonal rainfall between northern and southern China at glacial–interglacial timescales appears to have existed since the late Early Pleistocene [105]. In Bose Basin, the analysis of clay minerals, iron oxides, and geochemical indicators of the Gaolingpo profile suggested that the regional climate was temperate to warm temperate and then became warmer and more humid with a warm temperate to subtropical climate during the late Early Pleistocene to late Middle Pleistocene (~1.4–0.3 Ma) [106]. The increasingly warm and wet conditions during paleosol formation were also revealed by the Dawang red earth sediments in the northern Bose Basin [107]. In Chongzuo City, stable carbon and oxygen isotope analyses of mammalian tooth enamel samples from two caves showed the regional climate became wetter from the Early Pleistocene to the early Late Pleistocene, which may be due to the intensified glaciations [108]. This is consistent with the palynological records from Queque Cave, which showed that since the late Early Pleistocene, there has been a shift from temperate to warm conditions (MAT of 11.3–15.4 °C; MAP of 601.1–1076.1 mm) to probably subtropical humid conditions (MAT of 12.6–18.6 °C; MAP of 784.7–1523.1 mm) [109].

Based on the paleoclimatic evidence, it seems that both temperature and precipitation in South China were notably higher than those in North China under the impact of the glacial–interglacial climatic oscillations, which would have had different impacts on early human evolution in China, e.g., [38–40].

### 2.3. The Chronological Sequence of Early Human Evolution in China

The specific information on the 95 sites of the Early and Middle Pleistocene in China is listed in Table 1. We collected as many qualified sites as possible on the premise of ensuring the accuracy and reliability of the data. Here, we will analyze the chronological sequence and the possible distribution of these hominin settlements within the four areas mentioned above.

**Table 1.** The main archaeological sites of the Early and Middle Pleistocene in China.

| No. | Site | Age (Ma) | Dating Method | Loess–Paleosol Period | MIS | Homo (Species) | References |
|---|---|---|---|---|---|---|---|
| 1 | Shanshenmiaozui | 1.77–1.05 | PM | S24–L12 | 63–30 | | [110] |
| 2 | Shangshazui | 1.7–1.6 | PM | L24–S23 | 60–55 | | [111] |
| 3 | Putaoyuan | 1.6–1.5 | PM | S23–S21 | 55–50 | | [112] |
| 4 | Shigou | 1.6 | PM | S23–S22 | 55–50 | | [37] |
| 5 | Lanpo | 1.6 | PM | S23–S23 | 55–50 | | [113] |
| 6 | Majuangou 1–3 | 1.66–1.55 | PM | L24–S22 | 58–53 | | [37,114] |
| 7 | Dachangliang | 1.36 | PM | S17 | 43 | | [115] |
| 8 | Xiaochangliang | 1.36 | PM | S17 | 43 | | [116] |
| 9 | Banshan | 1.32 | PM | S17 | 41 | | [117] |
| 10 | Nanshanbian | 1.3 | PM | S16 | 40 | | [118] |
| 11 | Feilang | 1.2 | PM | S13 | 36 | | [119] |

**Table 1.** *Cont*.

| No. | Site | Age (Ma) | Dating Method | Loess–Paleosol Period | MIS | Homo (Species) | References |
|---|---|---|---|---|---|---|---|
| 12 | Madigou | 1.2 | PM | S13 | 36 | | [120] |
| 13 | Cenjiawan | 1.1 | PM | L12 | 32 | | [121] |
| 14 | Donggutuo | 1.1 | ESR | L12 | 32 | | [122] |
| 15 | Huojiadi | ~1 | PM | S11 | 28 | | [123] |
| 16 | Maliang | 0.78 | ESR | S7 | 19 | | [122] |
| 17 | Zhoukoudian locality-1 | 0.77; 0.55–0.3 | $^{26}$Al/$^{10}$Be; ESR/U | S7 | 19 | *H. erectus* | [124] |
| 18 | Zhuwobao | ~0.5 | ESR | S5 | 13 | | [125] |
| 19 | Hougou | 0.395 | PM | S4 | 11 | | [126] |
| 20 | Dongpo | 0.321 | ESR | S3 | 9 | | [127] |
| 21 | Motianling | 0.315 | OSL | S3 | 9 | | [128] |
| 22 | Queergou | 0.268 | OSL | L3 | 8 | | [128] |
| 23 | Shangchen | 2.12–1.26 | PM | S27–S15 | 80–38 | | [129] |
| 24 | Gongwangling | 1.63 | PM | L24 | 57 | *H. erectus* | [70] |
| 25 | Xihoudu | 1.27;1.4 | PM;$^{26}$Al/$^{10}$Be | S15 | 39 | | [130,131] |
| 26 | Longgangsi Locality 1–4 | 1.27–0.58 | post-IR IRSL; PM | S15–S5 | 39–15 | | [132,133] |
| 27 | Meipu | ~0.99–0.78 | BS; PM | S9–S7 | 28–19 | *H. erectus* | [134] |
| 28 | Yuelianghu | ~0.99–0.82 | PM; SC | S9–S8 | 28–21 | | [135] |
| 29 | Yunxian Man | 0.89–0.71 | PM | L9–S6 | 22–17 | *H. erectus* | [136] |
| 30 | Guanmenyan | 0.82–0.79 | PM; SC | L8 | 21–19 | | [135] |
| 31 | Shangbaichuan | 0.78 | PM | S7 | 19 | | [137] |
| 32 | Bailongdong Cave | 0.76 | $^{26}$Al/$^{10}$Be | S7 | 18 | *H. erectus* | [138] |
| 33 | Miaokou | 0.7–0.6 | PM; OSL | S7–S5 | 17–15 | | [139] |
| 35 | Chenjiawo | 0.65 | PM | L6 | 16 | *H. erectus* | [37] |
| 34 | Liuwan | ~0.6 | OSL; PM | S5 | 15 | | [140] |
| 36 | Shuangshu | 0.65–0.52 | ESR | L6–S5 | 16–13 | | [141] |
| 37 | Qiaojiayao | 0.62–0.6 | OSL; PM | S5 | 15 | | [142] |
| 38 | Yaochangwan | 0.58–0.47 | TT-OSL; SC | S5–L5 | 15–12 | | [132] |
| 39 | Jiuchang | 0.58–0.47 | post-IR IRSL; PM | S5–L5 | 15–12 | | [36] |
| 40 | Luojiacun | 0.58–0.47 | post-IR, IRSL | S5–L5 | 15–12 | | [132] |
| 41 | Wujiagou | 0.58–0.47 | $^{26}$Al/$^{10}$Be; PM | S5–L5 | 15–12 | | [36] |
| 42 | Pengjiahe | 0.58–0.47 | SC | S5–L5 | 15–12 | | [132] |
| 43 | Longyadong Cave | 0.41–0.26 | TT-OSL | S4–L3 | 11–8 | *H. erectus* | [143] |
| 44 | Jijiawan | 0.4–0.1 | OSL; post-IR IRSL; PM | S4–L2 | 11–5 | | [144] |
| 45 | Maling 2A | 0.39–0.22 | OSL | S4–S2 | 11–7 | | [145] |
| 46 | Dali Man | 0.27–0.26 | post-IR IRSL | L3 | 8 | Archaic *H. sapiens* | [146] |
| 47 | Zhoupo | 0.25–0.18 | TL | S2–L2 | 8–6 | | [147] |
| 48 | Dingcun | 0.21–0.16 | U | S2–L2 | 7–6 | | [148] |
| 49 | Yujiashan | 0.2 | U | S2 | 7 | | [149] |
| 50 | Beiyao | 0.2–0.07 | OSL | S2–L2 | 7–4 | | [150] |
| 51 | Yuling Man | 0.156 | U | L2 | 6 | Homo? | [151] |
| 52 | Renzidong cave | 1.24–1.03; 2.5–2.2 | ESR; PM | S15–S10 | 37–29 | | [152,153] |
| 53 | Qiliting Lower layer | 0.94–0.89 | PM | L9 | 25–22 | | [154] |
| | Qiliting Upper layer | 0.3–0.12 | PM | S3–L2 | 8–5 | | [154] |
| 54 | Chenshan | 0.82–0.13 | ESR | S8–L2 | 21–6 | | [155] |
| 55 | Yangshan | 0.73–0.4 | ESR | L7–S4 | 18–11 | | [156] |
| 56 | Maozhushan | ~0.6 | ESR | S5 | 15 | | [157] |
| 57 | Heshangdun | 0.5–0.13 | PM | L5–L2 | 13–6 | | [158] |
| 58 | Fangniushan Layer 8 | 0.45–0.21 | ESR | L5–S2 | 12–7 | | [159] |
| 59 | Hexian Man | 0.412 | ESR/U | S4 | 11 | *H. erectus* | [64] |
| 60 | Chaoxian Man/Yinshan | 0.36–0.31 | U | L4–S3 | 10–9 | Archaic *H. sapiens* | [160] |
| 61 | Nanjing Man | 0.35; 0.62–0.58 | ESR; U | L4 | 10 | *H. erectus* | [66,161] |
| 62 | Hualong Cave | 0.331–0.275 | U | S3–L3 | 9–8 | *H. erectus* | [162] |
| 63 | Lianhuadong Cave | 0.3–0.1 | U | S3–S1 | 8–5 | Archaic *H. sapiens?* | [163] |
| 64 | Shilongtou layer 1 | ~0.28 | U | L3 | 8 | | [164] |

**Table 1.** *Cont.*

| No. | Site | Age (Ma) | Dating Method | Loess–Paleosol Period | MIS | Homo (Species) | References |
|-----|------|----------|---------------|----------------------|-----|----------------|------------|
| 65 | Longgupo | 2.48–2.2 | PM; ESR/U | L32–S29 | 98–84 | Ape | [165] |
| 66 | Mohui Cave | 1.7–1.3 | ESR/U | L24–S16 | 60–40 | *H. erectus* | [166,167] |
| 67 | Yuanmou Man | 1.7; 1.54 | PM; $^{26}$Al/$^{10}$Be | L24 | 60 | *H. erectus* | [67,168] |
| 68 | Yangwu | 0.8 | $^{40}$Ar/$^{39}$Ar; SC | L8 | 20 | | [169] |
| 69 | Baigu | 0.8 | $^{40}$Ar/$^{39}$Ar; SC | L8 | 20 | | [169] |
| 70 | Fengshudao | 0.8 | $^{40}$Ar/$^{39}$Ar; SC | L8 | 20 | | [170] |
| 71 | Nanbanshan/Da Mei | 0.8 | $^{40}$Ar/$^{39}$Ar; SC | L8 | 20 | | [171] |
| 72 | Nalai | 0.8 | $^{40}$Ar/$^{39}$Ar; SC | L8 | 20 | | [172] |
| 73 | Baifeng | 0.8 | $^{40}$Ar/$^{39}$Ar; SC | L8 | 20 | | [173] |
| 74 | Gaolingpo Lower layer | 0.8 | $^{40}$Ar/$^{39}$Ar; SC | L8 | 20 | | [174] |
| 75 | Liuhuaishan | 0.8 | $^{40}$Ar/$^{39}$Ar; SC | L8 | 20 | | [175] |
| 76 | Panlong Cave | >0.441 | U | L5 | 12 | Archaic *H. sapiens* | [176] |
| 77 | Baidu | 0.4–0.3 | SC | S4–S3 | 11–9 | | [177] |
| 78 | Yumi Cave | 0.4–0.008 | U | S4–S1 | 11–4 | | [178] |
| 79 | Panxiandadong Cave | 0.3–0.12 | OSL | S3–L2 | 8–5 | | [179] |
| 80 | Maba Man | ~0.3 | U | S3 | 8 | Archaic *H. sapiens* | [69] |
| 81 | Zhangkou Cave | ~0.3 | U | S3 | 8 | Archaic *H. sapiens* | [180] |
| 82 | Yanhui Cave | ~0.24–0.21 | U | S2 | 7 | *H. erectus* | [181] |
| 83 | Xiaohui Cave | 0.23–0.13 | U | S2–L2 | 7–6 | Archaic *H. sapiens* | [182] |
| 84 | Lingfeng Cave | ~0.2 | U | S2 | 7 | | [183] |
| 85 | Changyang Man | 0.20–0.14 | U | S2–L2 | 7–6 | Archaic *H. sapiens* | [184] |
| 86 | Xinglong Cave | 0.15–0.12 | U | L2 | 6–5 | | [185] |
| 87 | Dongzhongyan | 0.148 | U | L2 | 6 | Archaic *H. sapiens* | [186] |
| 88 | Ranjialukou | 0.143 | OSL | L2 | 6 | | [187] |
| 89 | Jiaojiedong | 0.175 | U | L2 | 6 | | [188] |
| 90 | Dayao | ~0.43 | OSL | L5 | 11–9 | | [189] |
| 91 | Jinniushan locality A | 0.23–0.2 | TL | S2 | 7 | Archaic *H. sapiens* | [190] |
| 92 | Miaohoushan | ~0.53 | U | S5 | 13 | Homo.? | [191] |
| 93 | Yiyuan Man | 0.42–0.32; 0.64 | ESR/U; $^{26}$Al/$^{10}$Be | L5–S3 | 11–9 | *H. erectus* | [192,193] |
| 94 | Baishiya Cave | 0.16 | U | L2 | 6 | | [194] |
| 95 | Yangshang | 0.22–0.1 | OSL | S2–L2 | 7–5 | | [195] |

Note: PM: paleomagnetism; ESR: electron spin resonance; $^{26}$Al/$^{10}$Be: $^{26}$Al/$^{10}$Be burial dating; ESR/U: the combined electron spin resonance and uranium series; U: uranium series or U-series; OSL: optically stimulated luminescence; post-IR IRSL: K-feldspar post-infrared and infrared stimulated luminescence; TT-OSL: thermally transferred optically stimulated luminescence; TL: thermoluminescence; BS: biostratigraphy; SC: stratigraphic correlation; MIS: marine isotope stage [8]. The corresponding loess–palaeosol periods were referenced by Ding et al. [33]. When multiple ages were present, the former was adopted for the analysis. Numbers (No.) 1–22 denote sites in NHW-ZKD, No. 23–51 denote sites in QMR, No. 52–64 denote sites in the lower reaches of YRV, and No. 65–88 denote sites in Southern China. The geographical coordinates and context (open-air or cave site) of the 95 sites can be obtained in the supplementary material (Table S1).

### 2.3.1. The Nihewan Basin

The Nihewan Basin, located on the eastern edge of the Loess Plateau at the junction with the Inner Mongolian Plateau at a latitude of about 40° N, is an important center of early human settlements in northern China [196]. The long and well-developed fluvial–lacustrine sediments (known as the Nihewan Formation) in the region are rich in excellently preserved Paleolithic sites from all stages of the Pleistocene, particularly in the Early Pleistocene [34,196]. The Early Pleistocene sites were characterized by a relatively complete chronological sequence and abundant small stone artifacts and are considered to be the birthplace of the traditional small tool industry in North China [196,197]. The main methods used for dating Early Pleistocene sites (before ~0.78 Ma) in the basin are paleomagnetism (PM) and electron spin resonance (ESR), which have resulted in a long-scale, largely continuous chronological sequence of sites from ~1.7–0.78 Ma, yet there is a significant chronological gap in the period of 1–0.8 Ma, e.g., [110–122]. The most widely accepted

age of hominin activity in the basin is about 1.7–1.6 Ma [114]. These early humans were probably hunter-gatherers who used simple Olduvai (Mode I) stone artifacts such as cores, flakes, etc. [198]. The number of Middle Pleistocene (~0.78–0.3 Ma) sites is relatively small, and most of them appeared in the late Middle Pleistocene and were dated by PM, ESR, and optically stimulated luminescence (OSL), e.g., [125–128]. During this period, the stone tool industry was still the traditional small stone tool industry in North China, but compared with the Early Pleistocene, changes in the characteristics of stone artifacts were mainly reflected in the wider selection of raw materials and the beginning of planning in the production [197].

The reason for the scarcity of Middle Pleistocene sites in the Nihewan Basin may be twofold: On the one hand, there are limitations in the application of dating methods [199]. Since most of the early sites in the region are buried in fluvial–lacustrine deposits, it is hard to apply the loess–paleosol sequence to define the chronology; and the U-series dating method cannot be used if no animal fossils are preserved in the Paleolithic sites of this period [40,199]. On the other hand, the environmental changes under the influence of the climate and the scarcity of resources may be one of the reasons that drove the early humans to abandon the region [36,37,40], since most of the Middle Pleistocene sites existed in the interglacial periods during which the paleosols were developed, e.g., [125–127]. It is noteworthy that there were hominin activities in the nearby Zhoukoudian area in Beijing during 0.77–0.30 Ma [124]. It is speculated that the gathering place of hominins in northern China may have shifted from the Nihewan Basin to the Zhoukoudian area during this period [39,40].

### 2.3.2. The QMR

Most Paleolithic sites in the QMR are buried in eolian deposits or river terrace deposits in intermountain basins, mainly in the Bahe River basin in the Northern QMR, the Hanjiang River basin in the Southern QMR, and the Nanluohe River basin in the Eastern QMR [36,132,200]. Since the Qinling Mountains hinder the majority of dust transportation from northwest to southeast China, there is a reduction in the deposition of loess in the QMR basins [35,201]. In comparison to the Loess Plateau, the loess deposits found in the QMR exhibit thinner layers, finer grain sizes, lower sedimentation rates, and a significantly reddish color [201,202]. However, it still maintains a correlation with the well-known Luochuan loess–paleosol sequence [200,201]. To date, more than 30 Paleolithic and hominin fossil sites or localities have been discovered in the Bahe River Valley of Lantian in North QMR [35,200]. Hundreds of Paleolithic sites or localities have been discovered in Hanzhong, Ankang, Yunxian, and Danjiangkou in the Hanjiang River Valley in the southern QMR [132,200]. In the Luonan basin of the East Qinling Mountains, more than 300 sites and localities have been discovered [198,202].

Over the past decade, our research team has made great progress in dating sites in the QMR and has dated dozens of sites by PM, $^{26}$Al/$^{10}$Be, OSL, and U-series methods, providing a solid foundation for the establishment of a continuous chronological sequence of the early human evolution in the QMR, e.g., [36,132,139]. There are relatively few records of hominin activities in the QMR during the Early Pleistocene. The regional lithic assemblage belongs to the Oldowan (Mode I) lithic industry, and it is dominated by cores, flakes, choppers, and simple retouched flake tools in this period [200]. Representative open-air sites include Gongwangling (~1.63 Ma) [70], Longgangsi Locality 3 (~1.2 Ma) [132,133], and the Yunxian Man Site (~0.9–0.7 Ma) [136]. In the Middle Pleistocene, the number of hominin sites in the QMR increased during ~0.78–0.5 Ma, e.g., [132,137–142]. During this period, the lithic assemblage was still relatively simple and dominated by large and small cores, flakes, and small retouched flake tools such as scrapers and points [200].

### 2.3.3. The Lower Reaches of the YRV

Paleolithic sites in the lower reaches of the YRV are concentrated in Jiangsu, Zhejiang, and Anhui provinces around 30° N. Although a large number of sites (such as the site

complex in the Shuiyangjiang River Basin in Anhui Province) have been continuously excavated in recent years [203], the number of sites with absolute chronological data is relatively small, totaling 13. The dating accuracy is low and controversial due to the use of palaeomagnetic dating, e.g., [154–156]. Most of the sites have a wide range of ages or intermittently span several climatic phases [40]. For example, the palaeomagnetic age of the Renzidong Cave based on the fauna indicates that the site is dated to 2.5–2.2 Ma [154], while the ESR age is about 1.24–1.03 Ma [155]. Therefore, the accuracy of the dates needs further investigation. There are numerous sites in this area dating from approximately 0.6–0.1 Ma, many of which contain hominin fossils that have been dated using U-series or ESR methods, including the Hexian Man site (~0.4 Ma), Chaoxian Man site (~0.36–0.31 Ma), Nanjing Man site (~0.35 Ma), and Hualong Cave (~0.3 Ma), e.g., [64,66,160–162].

### 2.3.4. The Southern China

Paleolithic sites in southern China are mostly cave sites, distributed between 22° N and 26° N and scattered from west to east in Yunnan, Guizhou, Guangxi, Guangdong, and other provinces [40]. The dating methods used for the Early Pleistocene sites are mainly PM, ESR/U, and $^{26}Al/^{10}Be$ [165–168]. Representative sites from this period include Longgupo (2.48–2.2 Ma), Yuanmou Man (~1.7 Ma), and Mohui Cave (1.7–1.3 Ma) [165–168]. Early Pleistocene sites before 1.7 Ma may not be Homo fossil sites [165]. There is a relatively long gap between ~1.7 Ma and ~0.4 Ma, with most sites dating from the end of the Middle Pleistocene to the Late Pleistocene, e.g., [176–187]. Numerous Middle Pleistocene sites in South China were dated by the U-series, primarily due to the exceptional preservation of animal fossils in cave deposits, e.g., [69,176,180–182,184,186]. Additionally, OSL dating has also been used for dating some Middle Pleistocene sites, e.g., [179,187].

It is noteworthy that a large number of Paleolithic sites have been found in the Bose Basin and the nearby small basins in Guangxi Province. Considering that only Yangwu, Baigu, Fengshudao, Damei, and other sites have reported absolute dates [169–175], the chronological framework presented here does not encompass all sites. Most of the Paleolithic sites in the Bose Basin are buried in the laterite fluvial deposits on terrace 4 (T4) of the Youjiang River [169]. Many sites contain handaxes (bifacial large cutting tools, LCTs) and have similar characteristics to stone artifacts that often coexist with Australasian tektites, e.g., [169–171]. According to the $^{40}Ar/^{39}Ar$ dating of the tektites and stratigraphic correlation, these sites can be dated to around 0.8 Ma, e.g., [169–175]. However, there were few sites before and after this period, which may be attributed to the climatic conditions in this region [40]. The warm climate and abundant rainfall often lead to substantial erosion of the sedimentary deposits, which will hinder the effectiveness of paleomagnetism, stratigraphic correlation, and other dating methods for the Early Pleistocene sites [199].

## 3. Results

*Spatio-Temporal Variations of Hominin Activity*

To investigate the spatial-temporal variations of hominin activity within long-term climatic oscillations, we incorporated the chronology of these sites into the framework of the marine oxygen isotopes and the loess–palaeosol sequence (Table 1). Variations in the content of benthic foraminiferal $^{18}O$ in deep-sea sediments provide a perfect proxy for global climate fluctuations [8]. Higher oxygen isotope values represent the cold glacial periods, while lower values represent the warmer interglacial periods. The typical loess–palaeosol sequence in central China is a high-quality record for understanding the variability of the East Asian monsoon and the glacial–interglacial cycles [33]. During the glacial period, the EAWM transported large amounts of dust, which was deposited on the Loess Plateau to form the loess (L) layer. During the interglacial period, the dust input decreased, and the EASM brought abundant rainfall, contributing to the formation of the paleosol (S) layer. For the past 1.8 Ma, the loess–palaeosol record can be well correlated with the marine record [33].

The 95 archaeological sites discussed in this paper are not all buried in loess–paleosol strata, but they are mostly located in monsoonal China (Figure 1). Therefore, the loess–paleosol stratigraphic sequences can provide a chronological and environmental benchmark for hominin evolution [40]. The variation in the number of sites in the four early human gathering areas across the loess–palaeosol climatic phases (~1.8–0.2 Ma, S25–S2) is shown in Figure 2. According to marine and Chinese Loess Plateau records, climatic changes occurred with an average frequency of 41 kyr during the Early Pleistocene. These shifts were generally less intense compared to the Middle Pleistocene period when the dominant cycle of 100 kyr emerged after ~0.7 Ma [8,33,89].

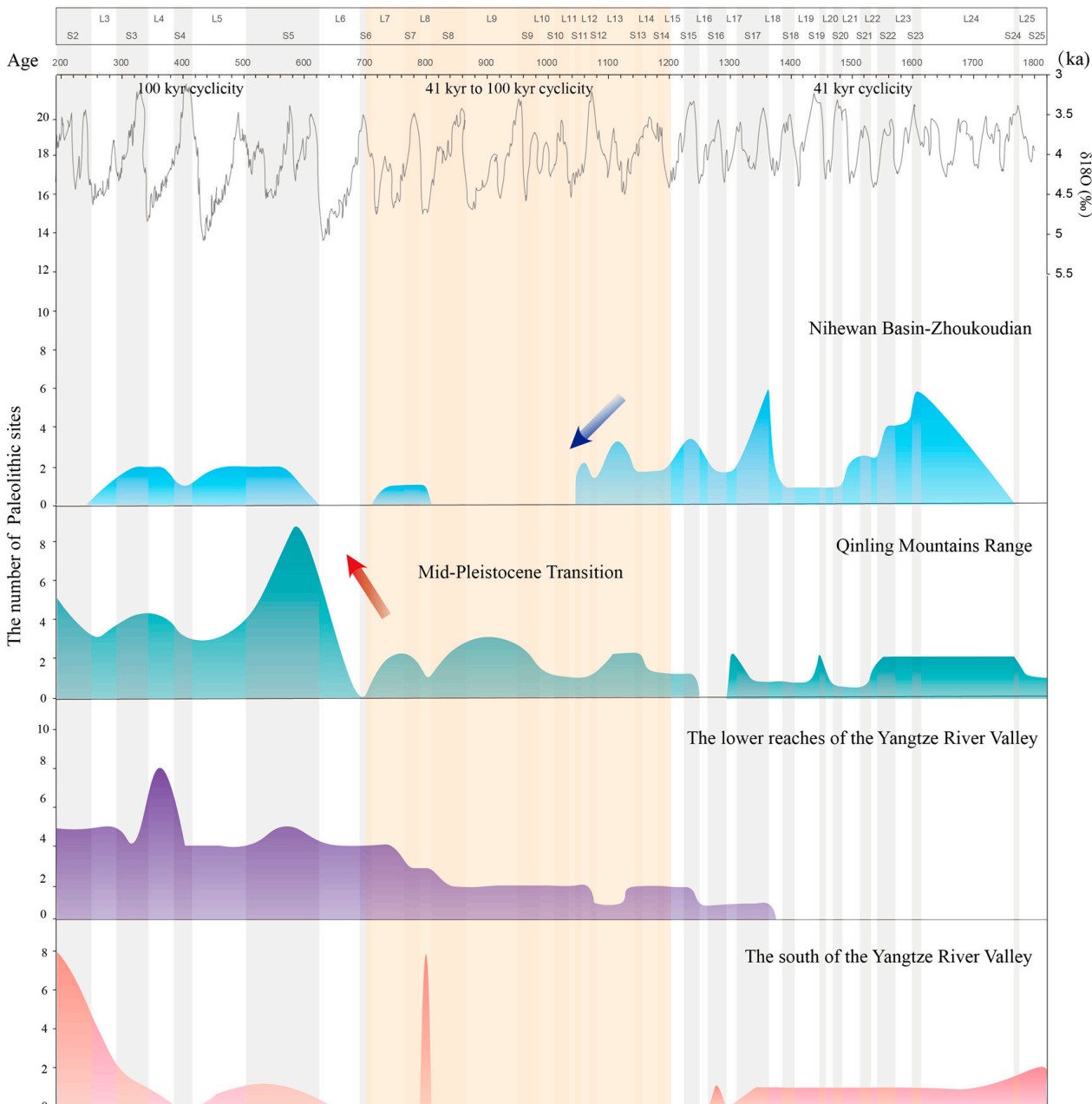

**Figure 2.** The tendencies of the number of Paleolithic sites in four early human gathering areas of China during the Early and Middle Pleistocene. Note: These four areas include Nihewan Basin-Zhoukoudian, Qinling Mountains Range, the lower reaches of the Yangtze River Valley, and the south of the Yangtze River Valley. We highlight the Mid-Pleistocene Transition with an orange vertical bar. The chronological data for all these sites are shown in Table 1.

In the Early Pleistocene, sites in NHW-ZKD occur continuously during the S24–S11 phase (MIS 63–28), with an obvious increase in numbers during the S23 interglacial phase (MIS 55). There are only a few sites in the QMR and even fewer sites in the lower Yangtze River Valley and Southern China. During the MPT, there was a decrease in the number of sites located in Northern China compared to earlier periods, and even a gap was observed during the L11–S8 phase (MIS 31–21). The number of sites in the QMR stabilizes at a low level, indicating continuity of human occupation. The number of sites in the lower reaches of the Yangtze River Valley remains relatively low but shows a gradual upward trend. Sites in southern China are mainly dated at about 0.8 Ma, e.g., [169–175], corresponding to the L8 glacial period (MIS 20) [8,33]. By the end of the MPT, the number of sites in NHW-ZKD was significantly lower compared with the early Pleistocene. During the MBE (~0.5–0.3 Ma), the site number of QMR showed an obvious increase, with a peak during the S5 interglacial period (MIS 15–13) [36,132,142]. Sites in the lower reaches of YRV are continuously present in the S15–L2 phase (MIS 37–5) [64,66,152–164], with a fluctuating upward trend in numbers. The number of sites in southern China increases to a greater extent during the S4–L2 phase (MIS 11–5) [69,177–187].

The spatial-temporal distribution of Paleolithic sites in the four early human gathering areas revealed that hominin activity was probably concentrated in northern China during the S24–S23 phase (MIS 63–55, ~1.7–1.6 Ma) [37,110–113]. It seems that the climate change during MPT (~1.2–0.7 Ma, S14–S6) significantly reduced the intensity of hominin activity in northern China, did not significantly change the intensity of hominin activity in QMR, and slightly increased the intensity of hominin activity in the lower reaches of the YRV and southern China [36,40]. Since about 0.6 Ma, the number of sites in the four regions has all increased, especially in the QMR and the lower reaches of the YRV, indicating that the intensity of hominin activities in the central and southern parts of China increased and probably tended to expand due to the warmer interglacial periods caused by MBE [38].

In summary, the number of early human sites in China increased significantly in the late Middle Pleistocene compared with earlier periods, indicating an overall increase in the intensity of hominin activity in all four regions. We deduced that the area with increased intensity of hominin activity gradually shifted southwards over time, as evidenced by the stepwise increase in the number of sites in southern China during the late Middle Pleistocene [37–40]. Given the limited number of sites with precise dates and the poor preservation of sites in certain areas, our preliminary findings aim to capture a general trend in the spatio-temporal variation of hominin activity. Our findings align with the conclusions drawn in previous investigations, e.g., [37–40]. Therefore, we cautiously speculate that northern China was no longer suitable for consistent hominin occupation due to the long-term climatic fluctuations, especially during the MPT. Early humans were forced to migrate to the more hospitable South.

## 4. Discussion

### 4.1. The Climate-Evolution Hypothesis

In Africa, numerous hypotheses have been proposed to clarify the link between climate change and hominin origin and evolution, such as the savannah hypothesis [16], the turnover pulse hypothesis [17], the variability selection hypothesis [18], the pulsed climate variability hypothesis [20], the accumulated plasticity hypothesis [21], the heterogeneity hypothesis [23], etc. Most of them attribute the environmental factors to extreme hydroclimate (wet and dry phases) and habitat heterogeneity [203,204]. An increasing number of hypotheses have emphasized the significance of orbital-scale climate oscillations, particularly during periods of high climate variability, e.g., [22,24]. It was suggested that climate change, whether long term or high frequency, influenced hominin evolution through its impact on the available resources in terrestrial ecosystems [2,205–207].

Early human survival in Europe was also strongly constrained by climatic conditions [25]. The Middle Pleistocene's frequent climatic fluctuations and glacial–interglacial effects led to a habitat-tracking movement referred to as "ebb and flow" [26], which means

populations in northern Europe periodically retreated to refuges in the south and expanded northwards again when environmental conditions improved [26]. Another "source and sink" model, proposed by Dennell et al., suggests that the "source" populations were in southern and south-eastern Europe, where the refugia allowed some populations to survive the glacial period [27]. The "sink" populations are in central and northern Europe, which were suitable for occupation only during the warm interglacial period. When environmental conditions deteriorated, many "sink" populations went extinct and/or retreated to southern refuges, where they mixed with local groups. Sink populations often relied on "source" populations for replenishment. This model explains the variable morphology of the Middle Pleistocene hominins in Europe [27,28]. The changes in the type, availability, and productivity of plant and animal resources were inferred to be the primary cause of population dynamics [27].

### 4.2. The Migration and Dispersal of Early Humans in China

Due to special latitudinal and topographic features, the occupation and movement of early humans in China were different from those in Africa and Europe. During the Pleistocene, much of the area in the east QMR was flat and likely formed a migration corridor for animals [55]. In the glacial period, open-steppe dwelling species and cold-adapted fauna migrated southward, even into the south of the QMR. During the interglacial period, species adapted to warm and wet conditions were capable of expanding their geographical ranges and enduring for extended periods in northern regions [57,208]. Similarly, hominins moved back and forth between the north and south of China [56]. One corridor over the Qinling Mountains led them to cross the mountains along low-lying river valleys. Another corridor skirts the QMR, allowing hominins and other mammals to move into the more open and flatter eastern region [56].

Recent developments in the population dynamics of early humans in China have been well supported by solid evidence. According to Bae et al.'s review of human fossils, *Homo erectus* occupied the region between the Yangtze and Yellow Rivers during the Middle Pleistocene [39]. As the climate became warmer by the end of the Middle Pleistocene, archaic *Homo sapiens* started to migrate northward [39]. When assessing the applicability of the "source and sink" model to Asia, Martinón-Torres et al. claimed that much of central and west Asia as well as the steppes of northern China were uninhabitable during the cold period of the late Middle Pleistocene, and North China may act as a "sink" for the hominin population [209]. In line with Sun et al.'s research [36], the southern Qinling Mountains may have served as a refuge during the glacial period and a source area for continuing human settlements [36]. This theory was backed up by further studies, e.g., [210,211]. Based on Yang et al.'s investigation, the climatic fluctuations during the MPT probably led to a southward migration of northern hominins [37,38]. In terms of northward dispersal, Zhu reported that early humans expanded from low latitudes, such as the subtropical Yuanmou and Bose Basin, through middle latitudes to high northern latitudes like the Nihewan Basin [32]. The LCTs that initially emerged at lower latitudes and then spread northward may also be a sign of the northward dispersal of early humans [37,212,213].

According to the above studies and our chronological record (Table 1 and Figure 2) [32, 36–40,56,57,209], we assume that the population dynamics of early humans in China throughout the Pleistocene manifested as follows (Figure 3). During the cold and dry glacial periods, hominins living in northern China left their original habitats and migrated southwards, surviving in the refuges in southern China, e.g., [36,37]. A few hominins may also adapt to the harsh environment and continue occupying the northern area, e.g., [120,196]. During the warm and wet interglacial period, hominins living in southern China expanded and gradually occupied habitats northwards. In brief, the early human settlement in China underwent a cyclical process of regional contraction and expansion, abandonment and recolonization, isolation, and integration [214].

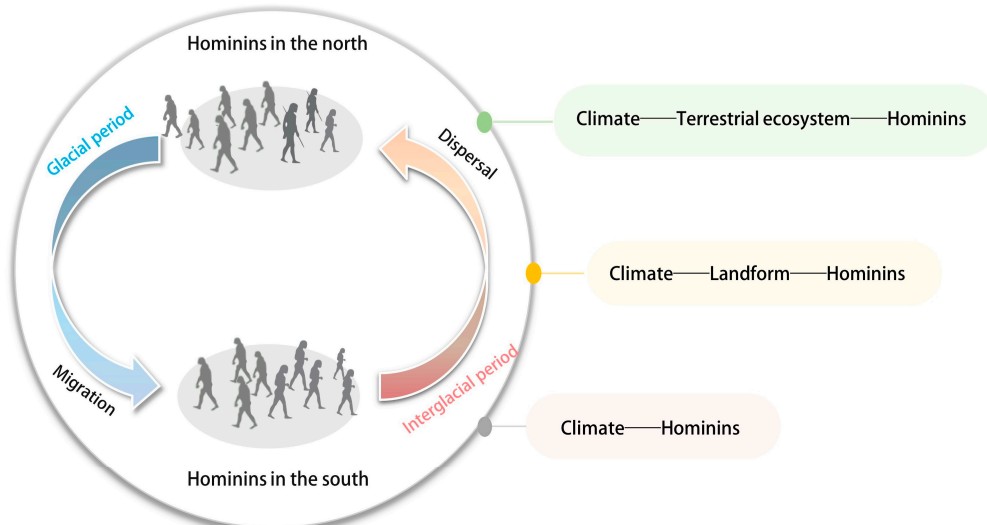

**Figure 3.** The migration and dispersal pattern of early humans in China and the potential driving mechanisms.

To be more specific, the QMR in central China was not an insurmountable barrier and did not impede the cultural exchanges between populations during the migration [36]. The QMR had mountains, valleys, intermountain basins, and river terraces as well as relatively abundant resources [200], serving as refugia for hominins that migrated between the southern and northern regions [36]. It is likely to be an important area connecting the Nihewan Basin, the lower reaches of the YRV, and southern China [135]. Dennell et al. also noted that the Yangtze River Valley may provide refugia for southern populations unable to occupy the north [215]. This bidirectional population movement was recorded in the demographic history of China in the last glacial cycle and supported by the analysis of ancient DNA [48,215]. Strategies combined with continuity and flexibility may characterize the response of most societies to climate change [216]. Note that the movement pattern is probably built over multiple climatic cycles. In other words, the brief duration of a single glacial–interglacial climate cycle may not significantly impact evolutionary history [217]. However, what is the driving mechanism behind this phenomenon? Here, we offer a preliminary explanation to clarify the lessons from history (Figure 3).

*4.3. Potential Driving Mechanisms of Population Dynamics in China*

4.3.1. Climate–Terrestrial Ecosystem–Hominins

The first explanation is that climate influenced the migration and dispersal of hominins by affecting plant and animal resources in terrestrial ecosystems, e.g., [27,36,207]. We believe that this driving mechanism is the most fundamental one, which can be summarized as "climate–terrestrial ecosystem–hominins". Prior to the advent of agricultural societies, hominins predominantly adopted a hunter-gatherer lifestyle, heavily relying on the available resources within terrestrial ecosystems, which constituted their primary habitats [218]. Early humans in China were no exception. During the Pleistocene, the fluctuating climatic cycles gave rise to diverse terrestrial ecosystems with different flora and fauna between north and south China [56,57], providing the basic environmental context for the local hunter-gatherer groups of *Homo erectus* and archaic *Homo sapiens* [31,37,38].

The quality of habitats is one of the most important factors in determining the migration of hunter-gatherers [219,220]. By altering available resources in terrestrial ecosystems, climate change affects both the size and geographic distribution of hunter-gatherer populations [219,221], which could be reflected in both individual and long-timescale climatic cycles. During the interglacial period, the EASM intensified and migrated northward, resulting in increased precipitation, accelerated soil development, increased vegetation cover, and more hospitable habitats in central and northern China. On the contrary, during

the glacial period, the EASM exhibited a weakened intensity and shifted towards southern latitudes. Consequently, the regional climate experienced arid and cold conditions, leading to an expansion of inhospitable habitats [87,211]. During the MPT, many regional records show a noticeable drying trend in inland Asia, with forest degradation and the expansion of grasslands and deserts in northern China, e.g., [222–224]. The total number of large mammalian species in North China decreased substantially during this period [224]. The original grassland mammalian fauna, e.g., *Equus sanmenniensis*, *Sus lydekkeri*, *Paracamelus gigas,* and *Cervus grayi*, remained the principal species during the MPT and adapted to the grazing and open habitats, while most of the carnivores of the Early Pleistocene, e.g., *Megantereon and Homotherium*, did not survive the MPT [224].

Forests have the highest species diversity among terrestrial ecosystems and are rich in plant foods, such as easily digestible leaves, bark, nuts, and fruits [224,225]. In contrast, grasslands provide a relatively limited food source for mammals, mainly stems and leaves that are difficult to chew and digest [224]. Take the Nihewan Basin as an example: The regional vegetation changed stepwise from the warm-temperate and temperate forests to the mixed forests and grassland [224], which led to the extinction of over 50% of the Early Pleistocene large mammals in North China and the southward movement of mammals [37,224,226]. It is suggested that *Homo erectus* tended to be large-prey hunters, e.g., [227–229]. At high latitudes, hunter-gatherer groups with low population densities are more dependent on animal-based foods [230]. Hunting and feeding on animals are vital to the overwintering of Nihewan hominins, e.g., [114,119]. The coexistence of faunal fossils with Paleolithic sites indicated that Nihewan hominins hunted and ate large herbivorous animals e.g., [110,114,119]. For instance, mammalian skeletal remains discovered at the Majuangou site exhibit distinct evidence of tool-induced damage, indicating that local hominins had already incorporated animal tissues into their diet by around 1.66 Ma [114]. Due to the impact of climate change, the Nihewan hominins would experience a scarcity of plant and animal resources, which may prompt the migration of some hominins toward southern regions [37].

In the QMR, ecological environments also changed in response to the glacial–interglacial cycles. However, there were always suitable habitats, even during glacial periods [36]. For example, Zhang et al. found that patches of woody vegetation persisted at the Longyadong Cave site in the Luonan Basin from 400–300 ka, which may provide diverse food resources for hominins surviving under climatic fluctuations [231]. Changes in terrestrial ecosystem structure and resource availability can be measured by vegetation productivity, which includes gross primary productivity (GPP) and net primary productivity (NPP) [36,232]. The reconstruction of GPP in QMR during the last glacial cycles suggested that the decrease in vegetation productivity was synchronized with a dramatic decrease in hominin activity [36].

Compared to the north, climatic fluctuations had relatively little impact on biodiversity in the YRV and South China, e.g., [233–235]. During the Pleistocene, the vegetation of the YRV was dominated by subtropical forests, and most of the large mammals were typical members of the "*Ailuropoda-Stegodon*" fauna, which included the giant panda, gibbons, orangutans, and the giant extinct ape *Gigantopithecus,* e.g., [234,235]. Generally, the number and abundance of edible plant species in low-latitude regions exceed those in high-latitude regions [55,56]. The exploitation of plant resources by early humans in southern China is also highlighted by some scholars [236,237]. Recently, Li et al. employed ecological modeling to analogize the paleoenvironment of the subtropical forests in Bose Basin [238]. They found that even during the extremely cold Last Glacial Maximum, the growing season of the vegetation still lasted for 11–12 months, indicating the area probably offered more stable and abundant plant resources for the LCT population (~0.8 Ma) [238].

4.3.2. Climate–Landform–Hominins

The second driving force behind early human migration may be the climate-induced earth surface processes, known as "Climate–Landform–Hominins". The fluctuating mon-

soon climate was a fundamental driver of the Earth's surface processes in East Asia [35]. In the loess–desert transition region of northern China, the movement of dunes in response to the EASM led directly to the occupation and abandonment of early human settlements [35,239,240]. Additionally, multilevel terraces were developed along many rivers in central and southern China in the Quaternary. Early human remains are widely distributed on these river terraces, such as the Hanjiang River, Nanluohe River, and Bahe River basins around the Qinling Mountains [200], the Youjiang River Basin in Southern China [169], etc. During the Pleistocene, tectonic uplift in these areas was relatively small, and the formation of river terraces was mainly regulated by the monsoon climate [241–243]. Increased climate fluctuations and monsoon precipitation during glacier–interglacial periods contributed to river erosion [244]. For the terraces of climatic origin, the depositional process occurs mainly during the glacial period, and downcutting occurs during the transition period between the glacial and interglacial periods [244,245]. We deduced that these river terraces may encourage hominin settlements.

### 4.3.3. Climate–Hominins

The direct effect of climate on hominins may also play a role in this story. The key factor lies in understanding the cold resistance of hominins [215]. There is no denying that hominin occupation in northern China usually occurred during warm and humid interglacial periods, e.g., [98,189]. Warm clothing, storable food, and the use of fire were important conditions for over-wintering [246]. The continuous occupation of northern latitudes by hominins was often accompanied by a variety of adaptive behaviors, such as the innovation of lithic technology in Nihewan Basin, e.g., [120,196,247], the emergence of Acheulean technology in Qinling Mountains [200] and traces of fire use at Zhoukoudian Locality [248,249]. In modern society, climatic conditions and other climate-related environmental factors play a crucial role in determining the livability of urban areas [250]. The most livable cities in China are mainly located in the south, such as Sanya in Hainan Province, Kunming in Yunnan Province, Xiamen in Fujian Province, and Zhuhai in Guangdong Province [251]. The preference for warm climates may be related to the genetic evolution of different hominin lineages [252,253].

According to the latest report, our Homo species favored areas with higher ecosystem diversity to seek abundant and diverse food [46]. We suggest that the movement pattern of early humans in China was essentially driven by the pursuit of subsistence resources [27,36,207], which was largely constrained by ecological factors in terrestrial ecosystems [254,255]. The main driving mechanism of the north–south migration and expansion of early humans in China is illustrated in Figure 4. At much lower latitudes, the southern region has a relatively warm and humid climate, a higher GPP, and abundant and diverse flora and fauna. It became both glacial refugia and an attraction for continuous hominin settlements. When climatic conditions ameliorated, the southern populations expanded, and some reoccupied the north. The preference for warm environments among hominins may be a biological factor. Additionally, the expansion of northern deserts and the formation of river terraces also played a role in the migration and dispersal of hominin populations.

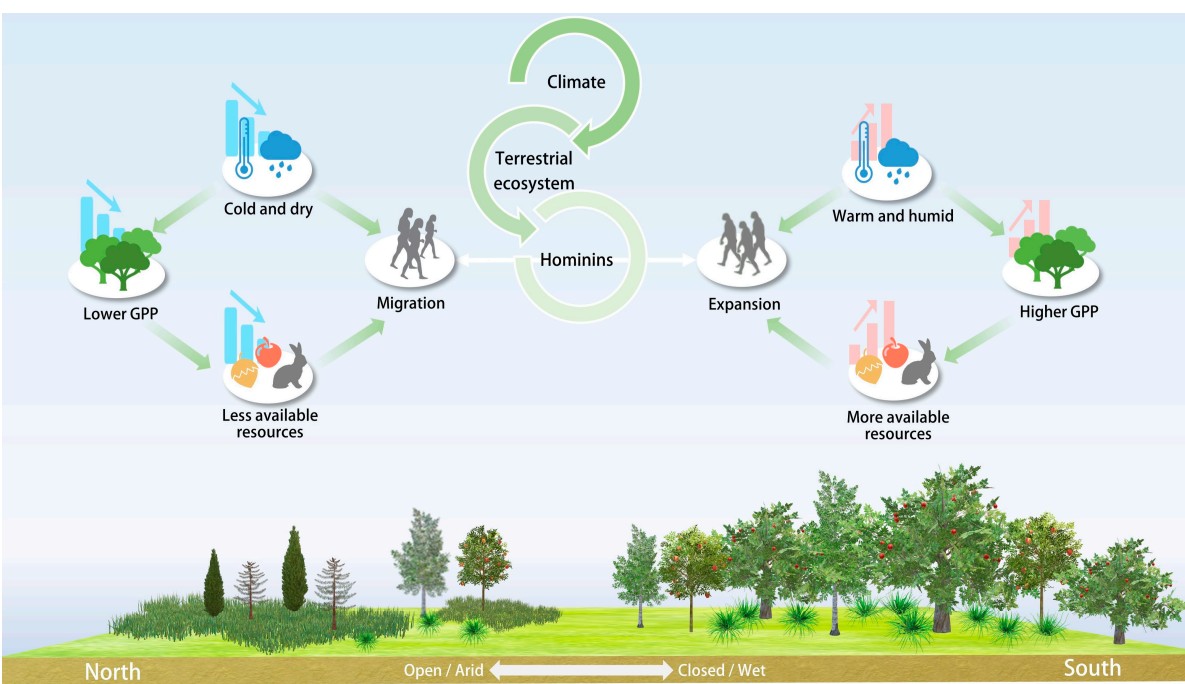

**Figure 4.** Main driving mechanisms of north–south migration and expansion of early humans in China. Note: Gross primary productivity (GPP) denotes the initial influx of energy and matter into terrestrial ecosystems. Higher GPP values indicate higher vegetation productivity and vice versa.

## 5. Conclusions

By combining paleoclimatic, chronological, and archaeological records, we hypothesized that the region of increased hominin activity in China gradually shifted southward during long-term glacial–interglacial climatic changes. The movement pattern of early humans in China may have been characterized by the southward migration during the cold/dry glacial period and the northward dispersal during the warm/humid interglacial period. The general trend of the population dynamics indicated that an adaptive strategy combined with continuity and flexibility against long-term climate change might have been adopted by Chinese hominins.

Among all the possible driving factors, the availability of resources in the terrestrial ecosystem was probably the most important one, whereas the landform and the cold resistance may play a subordinate role in shaping hominin evolution. Our findings are preliminary due to the limited number of Early to Middle Pleistocene sites with numerical dates. The knowledge about the population dynamics of Chinese hominins still needs to be refined by multidisciplinary studies, especially high-quality palaeoenvironmental and chronological datasets.

**Supplementary Materials:** The following supporting information can be downloaded at: https://www.mdpi.com/article/10.3390/land12091683/s1, Table S1: Specific data for the 95 archaeological sites.

**Author Contributions:** Conceptualization, X.S. and Z.Q.; data analysis, Z.Q.; writing—original draft preparation, Z.Q.; visualization, Z.Q.; writing—review and editing, X.S.; project administration, X.S. All authors have read and agreed to the published version of the manuscript.

**Funding:** This research was supported by the National Natural Science Foundation of China (Grant No. 41972185).

**Data Availability Statement:** The data used to support the findings of this study are available upon request.

**Acknowledgments:** We thank Lu Ying for her generous help in sorting out the previous data and designing Figure 2. We are sincerely grateful to editors and referees for their valuable comments and suggestions.

**Conflicts of Interest:** The authors declare no conflict of interest.

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
