# Peer review of "Glacial–Interglacial Cycles and Early Human Evolution in China"

_land, doi:10.3390/land12091683_

Round 1
Reviewer 1 Report (Previous Reviewer 1)
I judged that improvements have been made in the matters pointed out in the previous peer review. In particular, the presentation of basic archaeological data is important. The explanation of the mechanism that caused the migration is tentative and needs to be reexamined with the accumulation of more data in the future, but I think that this paper presents issues that require further discussion resonably.
Author Response
Thank you again for taking your valuable time to review our article. It is your suggestions that make our articles more rigorous and scientific. We will pay more attention to addressing the issues presented in this paper in our future work.
Reviewer 2 Report (Previous Reviewer 2)
Thank you for accepting all revisions and improving the references
Author Response
We sincerely thank you for your valuable suggestions, which made our article more complete and rigorous. Best wishes!

Reviewer 3 Report (New Reviewer)
The article is focused on hominin evolution and the Paleoenvironment, an important area of study that has been missing from academic discussions and research. An informed understanding of diverse terrestrial ecosystems during the Pleistocene, glacial and interglacial environments and hominin evolution is pivotal to informing our understanding of hominin evolution. Thus articles and research that address research on hominin evolution and inform our understanding of hominin subsistence and survival strategies in Paleoenvironments during the Pleistocene are a valuable addition to discussions of hominin and human evolution.
Line 17- A part of hominin groups- syntax- replace “part” to clarify this sentence
Line 17- northern hominin groups ? syntax- edit this sentence, we do not know if there were northern and southern communities of hominins, there were hominins, they most likely migrated during glacial-interglacial cycles. Perhaps clarify by stating something like hominin groups living in northern areas of the area we know today as Asia, migrated to
southern areas during glacial periods.
Line 18 The trend of southward migration became increasingly obvious over time. How did this become more obvious over time, is this known from the archaeological record?
Line 32 Old World Do not use colonial language. There is only one world, the entire world is old. Old World and New World are terminologies used by Western Eurocentric academics to create a view of the so called New World as less civilized and infantile in time and evolution. You can use the terms Eastern and Western Hemsiphere to discern these continental areas.
Line 43 paleoenvironmental information change this to paleoenvironmental data or evidence of ……..
Line 47 Concluded?, change conclude to perhaps discussed
Line 62 concerned by archaeologists perhaps say- of concern to archaeologists…..
Line 62 under the long term. Change to during the long term
Line 104 The early humans, delete The
Line 193 under mild climate – perhaps change to during times of milder climates
Line 204 from two fossil caves, edit- do you mean from two caves that contained fossils ?
Line 217 we try to collect- we collected
Line 290- In this period. You mean during this period?
Line 309 The Paleolithic sites. Delete The and change to Paleolithic sites
In the English language “Use “the” with any noun when the meaning is specific; for example, when the noun names the only one (or one) of a kind. Adam was the first man (the only “first man”). New York is the largest city in the United States (only one city can be “the largest”).
Line 317 U-series dating has been widely used since the Middle Pleistocene- syntax- edit this sentence it reads as though someone was using U-series dating during the Middle Pleistocene.
Line 347 Middle Pleistocene in this paper Do you mean to d-say discussed in this paper?
Line 410 influenced hominin origins and evolution. Delete origins
Line 514 Forest has the highest species diversity among terrestrial ecosystems. Do you mean one forest or many forest areas ?
Attention and editing required see above
Author Response
Point 1: Line 17- A part of hominin groups- syntax- replace “part” to clarify this sentence
Response 1: Done. We accept your comments.
Point 2: Line 17- northern hominin groups ? syntax- edit this sentence, we do not know if there were northern and southern communities of hominins, there were hominins, they most likely migrated during glacial-interglacial cycles. Perhaps clarify by stating something like hominin groups living in northern areas of the area we know today as Asia, migrated to southern areas during glacial periods.
Response 2: Thank you for your advice! We changed the whole sentence as” During glacial periods, hominins living in North China migrated to southern areas, while interglacial periods witnessed the northward expansion of hominins inhabiting South China.”
Point 3: Line 18 The trend of southward migration became increasingly obvious over time. How did this become more obvious over time, is this known from the archaeological record?
Response 3: Yes.As mentioned in Line 394-397 ”the area with increased intensity of hominin activity gradually shifted southwards over time, as evidenced by the stepwise increase in the number of sites in southern China during the late Middle Pleistocene”.We added the speculation in Line 467-469: “Based on the chronological and archaeological records (Figure 2), the north-to-south migration became increasingly conspicuous during the long-term glacial-interglacial shifts”. Since this finding is not emphasized in this paper, we deleted the sentense in abstract.
Point 4: Line 32 Old World Do not use colonial language. There is only one world, the entire world is old. Old World and New World are terminologies used by Western Eurocentric academics to create a view of the so called New World as less civilized and infantile in time and evolution. You can use the terms Eastern and Western Hemsiphere to discern these continental areas.
Response 4: Thank your for point out the important problem! We changed “the Old World” as “the Eastern Hemsiphere”.
Point 5: Line 43 paleoenvironmental information change this to paleoenvironmental data or evidence of ……..
Response 5: We changed “paleoenvironmental information” as “paleoenvironmental data”.Thank you for your suggestion.
Point 6: Line 47 Concluded?, change conclude to perhaps discussed
Response 6: Thanks. We accept your advice.
Point 7: Line 62 concerned by archaeologists perhaps say- of concern to archaeologists…..
Response 7: Thanks. We accept your advice.
Point 8: Line 62 under the long term. Change to during the long term
Response 8: Done.Thank you for your suggestion.
Point 9: Line 104 The early humans, delete The
Response 9: Thanks. We accept your suggestion.
Point 10: Line 193 under mild climate – perhaps change to during times of milder climates
Response 10: Thank you for your suggestion. We accept your advice.
Point 11: Line 204 from two fossil caves, edit- do you mean from two caves that contained fossils ?
Response 11: Yes,we changed as “from two caves”.
Point 12: Line 217 we try to collect- we collected
Response 12: We accept your advice.
Point 13: Line 290- In this period. You mean during this period?
Response 13: Yes,we changed as “during this period”.
Point 14: Line 309 The Paleolithic sites. Delete The and change to Paleolithic sites
In the English language “Use “the” with any noun when the meaning is specific; for example, when the noun names the only one (or one) of a kind. Adam was the first man (the only “first man”). New York is the largest city in the United States (only one city can be “the largest”).
Response 14: Thank you for helping us detect this problem! We examined the full text and made relevant changes.
Point 15: Line 317 U-series dating has been widely used since the Middle Pleistocene- syntax- edit this sentence it reads as though someone was using U-series dating during the Middle Pleistocene.
Response 15: We edited this sentence as “Numerous Middle Pleistocene sites in South China were dated by U-series”. Thank you!
Point 16: Line 347 Middle Pleistocene in this paper Do you mean to d-say discussed in this paper?
Response 16: Yes, we changed the sentence as “The 95 archaeological sites discussed in this paper”.
Point 17: Line 410 influenced hominin origins and evolution. Delete origins
Response 17: We accept your advice.
Point 18: Line 514 Forest has the highest species diversity among terrestrial ecosystems. Do you mean one forest or many forest areas ?
Response 18: We mean forests in the broad sense, therefore we changed the sentense as “Forests have….”.Thank you for pointing out the problem.

This manuscript is a resubmission of an earlier submission. The following is a list of the peer review reports and author responses from that submission.
Round 1
Reviewer 1 Report
This paper focuses on the relationship between climate change and hominin population dynamics during the Early and Middle Pleistocene in China, particularly how changes in climate environment corresponded to hominin occupations. The results of the analyses regarding the archaeological sites in several regions in China during the Paleolithic period revealed that the increase and decrease in the number of sites generally corresponded to the changes in the climate environment. From this, they found a pattern in which hominins moved geographically northward during warm periods and southward during cold periods. Based on this analysis, they propose the hypothesis that hominins during the Early and Middle Pleistocene were shifting their geographic habitats in response to changes in flora and fauna associated with changes in temperature.
The correspondence between environmental changes and hominin habitat that this paper seeks to argue is an important question for understanding human evolution in the Early and Middle Pleistocene. The claims that the authors attempt to make in this paper are clearly stated. It is unfortunate that, despite the importance of the hypothesis, we are unable to test its validity because of several problems identified in this paper.
(1) This paper examines 95 archaeological sites in four regions to investigate hominin population dynamics. However, the disclosure of data on the sites analyzed and their chronology is insufficient, and the reader cannot verify the validity of the results of the analysis. A list of the targeted sites and their estimated ages should be presented in this paper. Although it is described how the age of each of the occupied sites was estimated as being based on correspondence with the loess-paleosol sequence, it is unclear whether the accuracy and reliability of the estimated ages in the four areas are comparable. The reviewer cannot accept that the dating of the archaeological sites is as accurate as the natural environmental variations revealed by the oxygen isotope dating. The correspondence with the radiometric dating data obtained at each archaeological site is also not described.
(2) The number of sites treated in each region ranges from 13 to 27, which is too small a sample size to understand the changes in human activity over 1.6 million years. Given that the preservation and discovery of archaeological sites would have been greatly affected by sedimentation and erosion, as well as by modern human development, it is difficult to assume that this small sample reflects overall trends without bias. The issue of the process of site formation is important in discussing changes in the number of archaeological sites. The fact that there is absolutely no mention in this paper of how this issue affects the interpretation of the data is a major problem.
(3) Regarding the mechanism that brought about changes in the dynamics of hominin populations, hypotheses are raised in Chapter 5. However, this chapter is limited to general explanations, and no novel hypothesis is proposed. An explanation that is relevant to the Early and Middle Pleistocene of China is required. In particular, it is necessary to explain what kinds of plants and animals migrated from north to south in response to changes in climate and temperature, and how these migrations were related to hominin resource use.
Reviewer 2 Report
Dear Authors,
I have made some comments to your manuscript that you can find in the attachment
